# Development of a proposed solutions box for improving nutritional behaviors of female school students: Protocol of a mixed-method study

Nasrin Mehrjoyan[1], Fatemeh Zarei[1]*, Fazlollah Ghofranipour[1], Fazlollah Ahmadi[2]

**1** Department of Health Education & Health Promotion, Faculty of Medical Sciences, Tarbiat Modares University, Tehran, Iran, **2** Department of Nursing, Faculty of Medical Sciences, Tarbiat Modares University.Tehran, Iran

* f.zarei@modares.ac.ir

## Abstract

### Aim

Adolescent girls in Iran face significant challenges related to poor nutritional behaviors, such as inadequate fruit and vegetable consumption, high intake of processed foods, and irregular meal patterns. These behaviors contribute to long-term health risks, including obesity and micronutrient deficiencies. 'The aim of this study is to develop an evidence-based, context-specific 'solution box' to improve the nutritional behaviors of female school students in Khuzestan province.

### Design

A mixed-method study using a sequential exploratory design (Creswell model).

### Methods

This research will be conducted in three distinct stages. **Stage 1:** Qualitative Phase –**Part A:** A qualitative study using directed content analysis based on Social Cognitive Theory will be conducted. This phase will involve focus groups and interviews with female students, parents, and stakeholders to identify barriers to healthy eating and propose context-specific solutions. **Part B**: The perceived solutions will be extracted and categorized based on their relevance and feasibility. **Stage** 2: Quantitative Phase – Part A: Extracted solutions will be prioritized using the nominal group technique. Part B: Prioritized solutions will be ranked using a pairwise comparison scoring method. **Stage 3**: Solutions Optimization – Part A: A matrix of solutions will be developed based on theoretical frameworks. Part B: An action plan for implementing the solutions will be created. Part C: The solutions will be visualized on a digital platform for accessibility and usability.

which permits unrestricted use, distribution, and reproduction in any medium, provided the original author and source are credited.

**Data availability statement:** No datasets were generated or analysed during the current study. All relevant data from this study will be made available upon study completion.

**Funding:** The author(s) received no specific funding for this work.

**Competing interests:** The authors declare that they have no competing interests.

**Abbreviation:** SCT, Social Cognitive Theory; NUTRIBOX, Box of Solutions for Nutritional Behaviors; AHP, Analytic Hierarchy Process.

## Results

Not applicable for the protocol study.

---

## Background

Proper nutrition plays a critical role in the health and well-being of individuals during adolescence, a developmental phase characterized by rapid physical and mental growth [1]. Research highlights that nutritional deficits and unhealthy eating habits established during this period can have long-term consequences, affecting health, growth, and overall development well into adulthood [2]. Adolescence, which begins in the second decade of life [3], requires special attention due to its unique developmental needs and rights [3]. During this transitional stage from childhood to adulthood, health behaviors are solidified, shaping lifelong well-being. Girls, in particular, have distinct nutritional requirements compared to boys, especially during critical life stages such as puberty, menstruation, and pregnancy [4]. Essential nutrients such as iron, calcium, and folate are crucial for their growth and reproductive health. However, many adolescent girls struggle to meet these nutritional needs due to poor dietary habits, food insecurity, and limited access to nutritious foods. Social and environmental factors significantly influence girls' eating behaviors and attitudes [5]. Peer pressure, media exposure [6] and cultural norms contribute to body image concerns [7], potentially leading to unhealthy dietary practices [8].

Understanding the interplay among four fundamental concepts—nutritional problems, nutritional disorders, eating habits, and nutritional behaviors—enhances comprehension of this complex issue. Nutritional problems encompass a range of conditions resulting from inadequate or imbalanced nutrition, adversely affecting an individual's health and well-being. These problems manifest in various forms, including nutrient deficiencies or excesses, improper dietary patterns, weight fluctuations, and appetite disturbances [9]. Nutritional disorders, a subset of nutritional problems, arise from insufficient or excessive nutrient intake. Factors contributing to these disorders include inadequate absorption, increased nutrient requirements, nutrient loss, poor utilization, infections, and malnutrition [10].

Eating habits refer to conscious and repetitive behaviors related to food choices, influenced by factors such as taste, convenience, and cultural norms. These habits can be either beneficial or harmful and are shaped by emotional states, stress, boredom, and social contexts. Modifying eating habits requires self-reflection, replacing unhealthy patterns, and reinforcing positive choices [11]. In contrast, nutritional behaviors encompass a broad spectrum of actions related to food selection, dietary patterns, and overall food intake. This term includes food preferences, dieting practices, and concerns such as obesity and eating disorders. Socioeconomic status, cultural context, social environment, psychological factors, and education all influence nutritional behaviors [12]. Eating behavior, distinct from eating habits and nutritional behaviors, involves the complex interplay of dietary choices, attitudes, beliefs, and patterns related to food consumption. While eating habits focus on repetitive actions

(e.g., meal timing and portion sizes) and nutritional behaviors emphasize the health-related aspects of food choices (e.g., fruit and vegetable consumption), eating behavior encompasses broader psychological and social factors. For example, eating behavior includes emotional eating, food preferences, and the influence of social norms on dietary decisions. A 2022 meta-analysis revealed that over one-fifth of children and adolescents worldwide exhibit symptoms of eating disorders, with 22.36% of adolescents across 16 countries engaging in unhealthy dietary behaviors, such as excessive consumption of processed foods and sugary beverages [13]. Additionally, research indicates that Iranian teenagers are increasingly adopting a Westernized dietary pattern, characterized by frequent consumption of fast food and processed items such as sausages, salami, and pizza [14]. A national study by Klishadi et al. further reported that Iranian students regularly consume sweets, salty snacks, and fast foods while their intake of fruits, vegetables, and dairy products remains below recommended levels, posing significant health risks [15].

Nutritional behavior, as defined by the Department of Nutritional Behavior [16], Hoffmann (2016) [16], Oltersdorf (1984) [17] encompasses all actions—whether planned, spontaneous, or habitual—taken by individuals or social groups to obtain, prepare, and consume food. This definition also includes behaviors related to food storage and disposal. The concept of nutritional behavior extends beyond individual food choices to include broader health, environmental, social, and economic implications across the entire food supply chain, from production to consumption. Nutritional behavior is a multidimensional concept influenced by various intersecting factors, including health, environment, economy, and society [18–22]. However, existing interventions often focus on isolated aspects, such as individual behavioral changes [23], specific dietary habits [24], or calorie intake [25], without addressing the broader, interconnected nature of nutritional behavior. This gap highlights the need for a comprehensive approach. To address this issue, our study aims to enhance the understanding of the complex nature of nutritional behavior and develop an integrated solution framework through a mixed-methods approach. Addressing the complexity of nutritional behaviors in this specific cultural context requires a methodology that can both explore nuanced local perspectives and systematically prioritize interventions. A single-method approach would be insufficient; a purely qualitative study could identify potential solutions but would lack a structured mechanism for prioritization, while a purely quantitative study would risk imposing pre-conceived solutions that are not relevant to the lived experiences of students in Khuzestan. Therefore, this study adopts a sequential exploratory mixed-methods design. The initial qualitative phase is essential to explore and identify culturally-specific barriers and stakeholder-proposed solutions. Subsequently, the quantitative phase is necessary to systematically rank and prioritize these qualitatively-derived solutions, building consensus and ensuring feasibility. This integration allows the qualitative findings to directly inform the quantitative instrument, resulting in a final 'solution box' that is both empirically robust and contextually grounded. Recognizing the multifaceted influences on nutritional behavior, we have chosen Social Cognitive Theory (SCT) as our theoretical framework. SCT emphasizes the role of social and environmental factors, along with individual cognitive processes, in shaping behavior, providing a robust foundation for designing effective interventions.

## Objectives

The objectives of this study are as follows:

## Main Objective

Development of a Proposed Solutions Box for Improving Nutritional Behaviors of Iranian Female School Students

## The Specific Objective of the First Phase (Qualitative Study)

- Explaining the preventive nutritional perception, experience and behaviors of female school students based on Social Cognitive Theory (SCT)

**The Specific Objective of the Second Phase (Quantitative Study)**

- Prioritizing of extracted solutions from phase one (the nominal group method)

- Sequencing prioritized solutions (the scoring technique in pairwise comparisons)

**The Specific Objective of The Third Phase of research (Solutions Optimization)**

- Creating Matrix for the Solutions

- Providing an Action Plan for fixed solution

- Visualization of Solutions Box

## Methods

### Study design

A mixed-methods study with a sequential exploratory design will be conducted, following Creswell's framework [26]. This design involves two distinct phases:

   **Phase 1 (Qualitative):** Semi-structured interviews and directed content analysis to explore the subjective experiences and contextual factors influencing nutritional behaviors.

   **Phase 2 (Quantitative):** Prioritization and ranking of solutions identified in Phase 1 using quantitative methods. The philosophical foundation of this research is pragmatism [27]. This approach benefits the study by prioritizing practical outcomes and enabling the integration of qualitative insights (e.g., cultural barriers, personal experiences) with quantitative data (e.g., solution prioritization) to develop context-specific, actionable strategies (the solution box). Pragmatism aligns with the study's goal of addressing real-world nutritional challenges through a problem-centered, flexible framework. In this way, high-quality and rich information can be obtained to explain the concept and dimensions of preventive nutritional behaviors from the experiences of participants. Therefore, after a comprehensive literature review data will be collected through semi-structured interviews according to four constructs of SCT and will be analyzed by directed content analysis. The steps of the study design have been illustrated in Fig 1 (Fig 1).

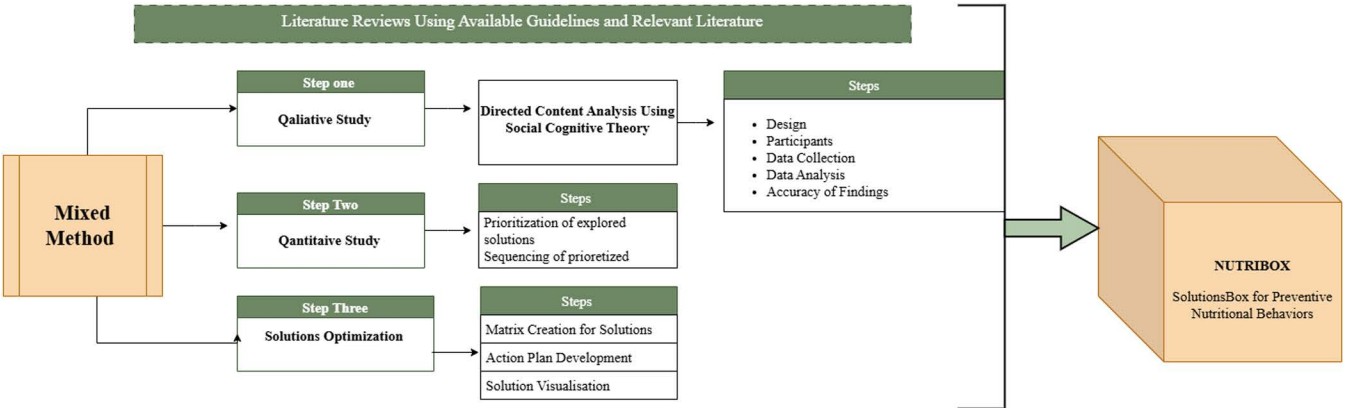

**Fig 1. Study Design of Development of a Proposed Solutions Box for Improving Nutritional Behaviors of Iranian Female School Students.**

## Literature review

The study is designed to identify and analyze the existing body of literature related to preventive nutritional perception and behaviors among female school students. This critical analysis is an essential part of the research process, as it helps establish what is already known about the topic and identifies gaps or further questions that the research aims to address. The literature review will focus on five key areas:

1. **Preventive Nutritional Behaviors:** We will search for relevant literature using keywords, concepts, theories, authors, and studies related to preventive nutritional behaviors in school students. This includes identifying factors influencing dietary habits and barriers to healthy eating.

2. **Best Practices and Interventions:** We will review literature on evidence-based interventions and best practices for improving nutritional behaviors in school students, with a focus on culturally appropriate strategies.

3. **Theory of Change:** We will explore literature on effective theories of behavior change, particularly those applicable to nutritional behaviors in adolescents.

4. **Social Cognitive Theory (SCT) Constructs:** We will identify and analyze predictive constructs of SCT (e.g., self-efficacy, observational learning) that are relevant to nutritional behaviors.

5. **National Guidelines:** We will review national guidelines and policies related to strategies and solutions for improving nutritional behaviors in school students, ensuring alignment with local context and priorities.

By synthesizing these areas, the literature review will provide a comprehensive foundation for the study and inform the development of the solution box.

The ultimate goal of this literature review is to identify gaps in current knowledge and inform the development of effective, context-specific solutions for improving nutritional behaviors among Iranian female school students. Furthermore, the outcomes of the literature review will elucidate which among the nine constructs of SCT yield predictive constructs of preventive nutritional behaviors. Consequently, predicated on these identified constructs derived from the comprehensive literature review, the guiding questions for the qualitative study will be formulated during the first phase of the research. This approach ensures a robust and informed foundation for the subsequent phases of the study.

## The first phase of the research: Qualitative study

In phase one, to align with the theoretical framework of Social Cognitive Theory (SCT), we selected directed content analysis as the most appropriate qualitative method for this study. SCT emphasizes the dynamic interplay between personal, behavioral, and environmental factors, and directed content analysis allows us to systematically analyze data through the lens of predefined SCT constructs (e.g., self-efficacy, observational learning, outcome expectations, and behavioral capability). This method ensures that the analysis remains anchored in SCT's theoretical principles while still capturing emergent themes from participants' lived experiences.

## Characteristics of the study population and participants

In this research, the primary participants (female school students) and secondary participants are stakeholders (school parents, parents of students, and health policy makers). The research will be selected with the purposeful sampling method, which is suitable for qualitative research [28]. In this method, the researcher is looking for people who are a rich source of information and can express themselves and their desire to participate in the research. Sampling and data collection will continue until the information saturation level is reached, that is, when the participants do not add new data to the previous data and the previous data are repeated [29].

## Method of data collection

The data collection method at this phase will be a semi-structured interview. At this stage, the first researcher (NM) as an experienced healthcare provider will be present in Secondary schools of Andimshek city, one of the big cities of Khuzestan province. Schools were selected using purposeful sampling to ensure representation across urban, semi-urban, and rural zones, as well as varying socioeconomic status (SES) levels. Participants (female students, parents, and stakeholders) were selected using stratified purposeful sampling to reflect diversity in age, education level, SES, and cultural background. Recruitment continued until data saturation was achieved. After introducing herself, explaining the importance and objectives of the research, and gaining the trust of the participants, written information which was approved by the ethics committee of Tarbiat Modares University and verbal consent will be obtained from the parents of students and the stakeholders, respectively. The time and place of the interview will be coordinated with them. Data will be collected using digital voice recorders and transcribed verbatim. Recordings will be stored on password-protected devices and encrypted to ensure confidentiality. The interview will be conducted individually and in a private and quiet environment desired and approved by the participants. The interview will begin with open-ended questions that allow the participants to fully describe their experiences of the phenomenon in question. At the beginning of the interview, questions will be asked to familiarize the researcher with the participants and also to create a sincere atmosphere, and as the interview proceeds, they will be guided to specialized questions in line with the research objectives (Table 1).

After the participant answers each of these questions, probing questions will be asked appropriately and the interviews will continue until the participants believe that they have no more information to provide. In case of fatigue or lack of data saturation of the data, the interview will continue in the next sessions with the same participant. At the end of the interview, the participants will be asked to state if there is anything else left. Then, while thanking the participant, the possibility of needing an interview or the next question is raised. In addition to audio recording, the researcher will record all emotions, facial changes, tone of voice, etc. during the interview with a pen and paper. After the completion of the interview, the recorded items will be played carefully as soon as possible. The original interviews will be kept in confidential folders. Analysis and initial coding of the data of each interview will be done before the next interview. After conducting the interview, the text will be presented to the supervisors and the necessary guidance will be obtained to improve the next interview process.

## Data analysis

To answer the research questions and analyze the data, the qualitative content analysis method will be used. To analyze data in the first step of phase one (qualitative study), the content analysis method will be used with a directed (deductive) approach. The analysis process will be done simultaneously and continuously with the data collection and based on the approach proposed by Shannon and Hsieh, through qualitative directed content analysis [28]. Directed content analysis, as described by Shannon and Hsieh, is a systematic approach used in qualitative research to interpret meaning from the content of text data. This method begins with a theory or relevant research findings that serve as guidance for initial codes. The process of directed content analysis can be summarized in the following steps:

—**Selection of Theory or Texts**: This initial stage encompasses the identification and examination of theories and texts pertinent to the phenomenon under study. The selection of an appropriate theory or text for the analysis of the phenomenon is a crucial part of this step.

—**Extraction of Conceptual Characteristics**: This step involves a thorough review of the chosen theory and related texts. The characteristics of the concepts and components of the theory, or the findings of the texts, are defined and delineated. These characteristics serve as vital information for subsequent analysis.

—**Matrix Construction**: A matrix is formulated wherein the concepts and dimensions identified in the previous step are arranged in columns and rows. This matrix acts as a reference framework for data analysis.

**Table 1. Interview guide questions based on the main constructs of the Social Cognitive Theory.**

**What are the perceptions and nutritional behaviors of adolescent girls to healthy eating?**

| Interviewee | Constructs | Interview Questions | Probing Questions |
|---|---|---|---|
| Adolescent Girls | **Self-Efficacy** | Do you easily eliminate unhealthy foods from your diet? And if not, what are the obstacles that prevent you from doing this? Can you share your experience in dealing with pressures and stress? How do you manage them? And in terms of nutrition, what experiences do you have? How do you control your nutritional behaviors in these situations? | Based on the answers: What do you mean by this? Could you explain more and give an example? |
| | **Outcome Expectations** | Have you ever noticed a change in your weight? Has your diet played a role in these weight changes? Have you experienced the negative effects of unhealthy nutrition on your body? What positive effects of healthy nutrition have you experienced in your diet? Have you had an experience when interacting with others that your dietary behaviors have been approved or rejected by them? How have your friends or classmates influenced your dietary choices? Have advertisements and media influenced your dietary choices? Have you had an experience of failure in making changes in your dietary pattern and how have you dealt with it? | Like what? Do you also have experience in this area? |
| | **Environment** | Do you have access to healthy foods and fresh ingredients in your living environment? Do your neighbors and local communities influence your food selection and consumption? Has access to nutritional information and awareness of health issues in your social environment caused a change in your dietary pattern? | Why? How? Can you explain more? The above questions can also be asked. |
| | **Self-Regulation** | Are you familiar with your body's needs and do you have a diet that matches these needs? Do you have successful experiences from your dietary pattern that you want to share? What have been your past experiences in goal setting in the field of nutrition? What problems have you encountered in goal setting in the field of nutrition? | The above questions can also be asked. Do you have an experience? Can you define it? |

**What are the perceptions and experiences of parents about the nutritional behaviors of their children to healthy eating?**

| Interviewee | Constructs | Interview Questions | Probing Questions |
|---|---|---|---|
| Parents | **Self-Efficacy** | Do you use any specific encouraging measures to promote healthy eating behaviors? How do you appreciate your children's positive actions in terms of nutrition? Do you use any specific encouraging measures to promote healthy eating behaviors? How can you provide support when your child is facing nutritional problems? | Based on the answers: What do you mean by this? Could you explain more and give an example? |
| | **Outcome Expectations** | How can you as parents help your child's physical health? What positive effects have you observed from a healthy diet in your child? Have you observed the experience of social disapproval of your child's eating behaviors? How can you as parents balance between the approval and motivation of your children for performing healthy eating behaviors? What suggestions do you have for strengthening parental social approval of dietary behaviors? | Like what? Do you also have experience in this area? |
| | **Environment** | Have you observed your child's experience of buying and choosing food items in stores? Does your child have experience in the process of preparing and provisioning food at home? Does the experience of social pressures (like appearance standards) affect your child's eating behaviors? Can parties and social activities influence your child's dietary patterns? | Why? How? Can you explain more? The above questions can also be asked. |

**What are the stakeholders' perceptions of dietary behaviors in adolescent girls to healthy eating**

| Interviewee | Constructs | Interview Questions | Probing Questions |
|---|---|---|---|
| Stakeholders | Self-Efficacy | What strategies have you adopted to encourage students to have healthy eating behaviors in the school environment? Has interaction with families been considered to promote healthy dietary patterns in school? Is there encouragement for students to make independent decisions about nutrition in school? Does the school collaborate with other institutions such as health centers, families, and community organizations? | Based on the answers: What do you mean by this? Could you explain more and give an example? |
| | Outcome Expectations | What experiences have you had in observing students' dietary patterns? Does the dietary pattern of students have an effect on their energy level and motivation in the educational environment? Has encouraging students' dietary behaviors affected school activities? Have you observed that students' positive or negative follow-up in terms of nutrition has affected their academic performance? Do you have experience that has helped to improve students' positive evaluation in terms of nutrition? | Like what? Do you also have experience in this area? |
| | Environment | Do students participate in decisions related to nutrition in school? What experience do you have about the impact of nutrition-related advertisements in school on students' dietary behaviors? How can the financial situation of students affect their way of nutrition? | Why? How? Can you explain more? The above questions can also be asked. |

—**Data Collection**: Data is collected through interviews, with an interview guide prepared based on the main concepts, characteristics of the concepts, and definitions presented in the matrix. The collected data serves as the primary source for further analysis [28]

—**Selection of Semantic Units and Code Extraction**: Semantic units are selected from the data and codes are extracted for them, guided by the main research question and the concepts defined in the theory. These codes function as key markers for the data.

—**Code Placement in the Matrix**: The extracted codes are positioned in the matrix prepared in step 3. This step facilitates the alignment of the data with the defined concepts and enables a more detailed analysis.

—**Operational Definitions for Main Concepts**: Operational definitions for each of the main concepts of the theory are provided by comparing the codes and results obtained from the data analysis with the theoretical definitions in the matrix. These definitions aid in the precise interpretation and scientific analysis of data [28].

It enables the researchers to examine the phenomena under scrutiny scientifically and in details for better comprehension. Besides, this approach may be used as a tool of shaping or amending predefined theories and concepts.

## Accuracy of the findings

To validate the findings of this research, Johnson's five criteria, which include credibility, confirmability, reliability, transferability, and authenticity, will be used [30]. Credibility: We will try to ensure the validity of the data, and enough time will be allocated to collect the data. There will be a long-term engagement with the data and immersion. Also, by combining data sources and theories, sampling from different comprehensive schools located in different zones of the city of Aandimeshk, the maximum diversity in sampling will be observed. Confirmability: For this reason, the researcher will try to describe all the stages of the research including data collection, analysis, and extraction of codes and classes in such a way that other people will be able to judge the correctness of the data by reading them. After writing the text, the co-researcher, who did not participate in the data collection stage, applies the necessary changes by comparing the text with the tape. To further confirm the validity of the content, several interviews will be coded and returned to the participants and the specialist for review. To ensure verifiability, the researcher will try not to interfere with his assumptions as much as possible in the process of data collection and writing. Reliability or Dependability: The researcher will ensure the reliability of the research by performing actions such as a review by the research group and analysis of observers in the 6-month reports. Transferability: We will try to select more diverse participants (students and parents) in terms of age, education, place of residence, and social, cultural, and economic status (maximum variance), and this process will continue until data saturation. Authenticity: This will be ensured by carrying out measures such as targeted sampling and the maximum diversity of participants (Table 2).

## The second phase of the research: Quantitative study

This step of the study design for the proposed solution box for improving the nutritional behaviors of female school students. This phase encapsulates the prioritization, weighting, and ranking of extracted context-based solutions from the qualitative phase. This phase aims to quantify the qualitative findings obtained from the first phase of the study.

**A: Prioritization of explored solutions.** In this phase, solutions derived from the initial qualitative phase will undergo prioritization. This prioritization will be conducted using the Nominal Group Technique (NGT), a structured group process designed for idea generation and prioritization. The implementation of the NGT is as follows:

The NGT has been widely recognized as an effective exploratory research method in scenarios that necessitate complex problem-solving and decision-making [31]. In this approach, group members independently pen down their solutions to the problem. Each member then independently and confidentially ranks these solutions. This method eliminates the phenomenon of social loafing, thereby encouraging more active participation from typically passive individuals. The nominal group will comprise experts in the fields of nutrition, education, health promotion, behavioral science, psychology,

**Table 2. Accuracy of the Data from female school students and stakeholders concerning selected SCT constructs in line with the Solutions Box development for Improving Nutritional Behaviors of Iranian Female School Students.**

| Criterion | Activities Used in Present Research |
|---|---|
| Credibility | • Sustained involvement and perpetual scrutiny<br>• Evaluation by a peer in the scientific community<br>• Conducting scholarly discussions with specialists who are not directly participating in the present research<br>• Assessment by contributors<br>• Cross-verification from multiple data origins:<br>  ◦ Interview with students<br>  ◦ Interview with experts in nutrition<br>  ◦ Interview with parents associated with the school<br>  ◦ Interview with parents of students<br>  ◦ Interview with policymakers |
| Confirmability | Detailed reporting of all work steps and receiving approval from participants who taking part in the research, members of the research team, and stakeholders |
| Reliability or Dependability | Continuous comparative analysis<br>Review by a colleague<br>Control by an external observer |
| Transferability | Choosing participants with maximum diversity |
| Authenticity | Each identified class should include at least one related document |
| | Targeted sampling |
| | Diversity of research participants for data collection and analysis |

and key individuals associated with adolescent nutritional behavior. The group size will range from 10 to 20 individuals. Solutions will be prioritized based on criteria such as availability, feasibility, and cost-effectiveness.

**B: Sequencing prioritized solutions.** The prioritized solutions will be ranked using a scoring technique based on pairwise comparisons. The primary objective of the Analytic Hierarchy Process (AHP) method is to select the optimal option based on various criteria through pairwise comparison. This technique is also employed for criteria weighting. The data collection tool in the AHP method is an expert questionnaire [32]. Pairwise comparison is utilized to determine the weight of the criteria and to rank the options [33] The questionnaire used for rank analysis and multi-criteria decision-making is referred to as an expert questionnaire. A pairwise comparison of elements is employed to prepare the expert questionnaire. A pairwise comparison matrix is prepared for each level of the hierarchy. For scoring, a nine-point scale is used. the scoring table will be done based on the nine-point scale

**The third phase of research: Solutions optimization**

**a) Creating a matrix for solutions.** This step involves the systematic organization of prioritized solutions into a matrix, which is a common tool in strategic planning and decision-making processes. In this phase, the solutions that have been prioritized and weighted in the preceding steps are represented in the form of a matrix. the solutions, which have been agreed upon by the experts under separate theoretical frameworks including the three strategies of the Ottawa Charter for Health Promotion [34], Social Determinants of Health [35], constructs of the SCT, Ecological focus [36], and the triple levels of prevention, will be organized into a matrix. This matrix serves as a comprehensive visual representation of the solutions, facilitating easier understanding and interpretation.

**Rationale for Compiling the Solution Matrix Based on Separate Theoretical Frameworks**

**1. Three Strategies of the Ottawa Charter in Health Promotion and Behavior Change Planning** Health promotion, now more than ever, is crucial in addressing public health issues. It has been increasingly acknowledged over the past

decades that biomedical interventions alone cannot ensure optimal health. The Ottawa Charter underscores the importance of addressing social, economic, and environmental determinants of health to achieve holistic well-being. Thus, achieving the highest possible health standard requires a comprehensive approach that transcends traditional medical care. This approach empowers individuals and communities to take health actions, strengthens public health leadership, promotes intersectoral action for healthy public policies, and builds sustainable health systems. Consequently, considering the strategies outlined in the Ottawa Charter can significantly enhance the planning of health behavior change interventions [37].

2. **Social Determinants of Health:** Considering social determinants of health in planning health behavior change interventions is essential for devising effective, equitable, and sustainable strategies. Addressing the root causes and underlying factors influencing health behaviors can significantly impact population health outcomes [38].

3. **Constructs of Social Cognitive Theory**: Theories can elucidate the structural and psychological aspects of behavior and guide the development and modification of health promotion and education efforts. Health behavior theories focus on behavior determinants at various levels – individual, interpersonal, group, organizational, and/or community. Integrating these theories into health behavior change intervention planning offers a systematic and evidence-based approach, increasing the likelihood of successful outcomes by addressing human behavior complexities and tailoring interventions to the target population's specific needs and contexts [39].

4. **Ecological Focus:** Effective health promotion programs typically operate at multiple levels, focusing not only on at-risk populations but also on environmental conditions that significantly influence health and health behavior. This perspective acknowledges multilevel behavior effects and emphasizes interconnected systems at individual, interpersonal, organizational, and societal levels. Understanding contextual factors and modifying environments support interventions tailored to behavior change complexity. Hygienic behavior is crucial at every societal level. However, achieving health promotion goals at any societal level necessitates behavioral changes among those controlling or influencing the desired health outcomes [40]

5. **Triple Levels of Prevention** Attention to the three levels of prevention – primary, secondary, and tertiary – is crucial in planning health behavior change. This approach enables a comprehensive strategy, targeting root causes with primary prevention, addressing early symptoms through secondary prevention, and mitigating the impact of existing conditions with tertiary prevention. Incorporating these prevention levels allows planners to proactively address health risks, prevent disease onset, and manage existing conditions, leading to a more precise and effective approach to behavior change [41].

   **b) Providing an Action Plan for a Fixed Solution.** This step involves the development of action plans for each solution, which is a key aspect of health promotion and prevention strategies. The action plan is a strategic instrument designed to delineate specific steps, tasks, and objectives necessary to accomplish a particular goal [42].The primary purpose of the action plan, based on the solutions agreed upon by the experts for improving the nutritional behaviors of adolescent girls, is to provide a roadmap. This roadmap guides individuals, teams, or organizations in effectively planning and implementing their goals.

   **c) Visualization of Solutions Box: NUTRIBOX.** The proposed solution box, termed NUTRIBOX, is a toolkit designed to improve the nutritional behaviors of Iranian female school students. It will contain evidence-based, context-specific strategies such as educational materials, interactive tools, and actionable plans tailored to the cultural and socioeconomic context of the target population. Developed through qualitative research using a directed content analysis approach guided by Social Cognitive Theory (SCT) constructs, NUTRIBOX is based on the experiences and insights of participants, including students, parents, stakeholders, and decision-makers.

   As the primary output of the study, NUTRIBOX will be presented as a digital platform (either an application or a website) to ensure accessibility and user-friendliness. This aligns with modern approaches to health promotion and

prevention. On the platform, the derived solutions will be detailed, and corresponding action plans will be visible and selectable by the user. By integrating SCT constructs (e.g., self-efficacy, observational learning, behavioral capability, and outcome expectations), NUTRIBOX addresses both individual and environmental factors influencing nutritional behaviors. For example, it may include:

- **Interactive video modules** featuring peer role models demonstrating healthy meal preparation (observational learning).

- **Self-assessment tools** to help students track their progress and build confidence in making healthy food choices (self-efficacy).

- **Skill-building workshops** on reading nutrition labels and planning balanced meals (behavioral capability).

- **Gamified challenges** where students earn rewards for achieving dietary goals, reinforcing positive outcomes (outcome expectations).

- **Community forums** where students, parents, and nutritionists can share experiences and support each other in adopting healthier habits (Social support).

In brief the final stage of this study focuses on translating the research findings into a practical and sustainable application. The "results" from the quantitative prioritization phase (Stage 2) will directly inform the creation of the final 'solution box.' This will not be a static report but a dynamic, user-centered digital toolkit designed for practical implementation by key community members.

The application will be developed as a web-based platform accessible to students, parents, and school health staff. The content will be tailored to each audience:

- **For Female Students:** The platform will feature interactive modules, culturally-appropriate healthy recipes, goal-setting worksheets, and short, engaging videos based on the prioritized solutions.

- **For Parents:** It will offer downloadable resources, such as fact sheets on adolescent nutrition, guides for healthy family meals, and strategies to support their children's dietary habits, all grounded in the solutions co-developed with the community.

- **For School Staff and Stakeholders:** The toolkit will provide a clear action plan, implementation guides for school-wide initiatives (e.g., healthy canteen policies), and ready-to-use educational materials for classroom integration.

By structuring the application this way, the study ensures that the empirically prioritized solutions are not merely identified but are transformed into an actionable resource that empowers the entire school community to foster healthier nutritional behaviors.

### Ethics approval and consent to participate

All methods were carried out following relevant guidelines and regulations (Helsinki Declaration of Ethical Principles for Medical Research and Ethical Research Guidelines Involving Children (ERIC) guidelines. Ethical approval was obtained from the Ethics committee of the faculty of medical sciences, at Tarbiat Modares University (IR.MODARES. REC.1402.247).

## Discussion

Existing research highlights the necessity of developing a comprehensive strategy to establish a solutions box for improving nutritional behaviors among female students. This demographic is particularly vulnerable compared to other groups, necessitating a nuanced understanding of the economic, social, and cultural factors influencing their dietary habits [43–45]

Food security—defined as consistent access to sufficient, safe, and nutritious food—is a crucial determinant of dietary patterns. In regions such as Khuzestan, economic disparities and environmental challenges (e.g., extreme heat and

humidity) can limit food availability and access, exacerbating food insecurity and contributing to unhealthy dietary habits [46]. For example, restricted access to fresh fruits and vegetables may lead to an increased reliance on processed or calorie-dense foods, elevating the risk of malnutrition and obesity [47]. Therefore, ensuring food security should be a foundational component of any intervention aimed at improving nutritional behaviors, particularly among vulnerable populations such as female students.

The proposed strategies for enhancing the nutritional behaviors of Iranian female school students, derived from the experiences of students, parents, stakeholders, and decision-makers, will be explored through a qualitative research approach using directed content analysis based on Social Cognitive Theory (SCT). SCT, widely applied in dietary interventions [48], posits that learning occurs in a social context, with behavior change influenced by environmental factors, personal attributes, and behavioral characteristics.

A comprehensive approach to promoting healthier dietary behaviors among students must incorporate multiple strategies, including educational, parental, policy-driven, environmental, peer-based, community, and technological interventions. School-based educational programs are instrumental in fostering students' understanding of nutrition and encouraging healthy eating habits [45]. Parental involvement is equally critical, as parents significantly shape their children's dietary behaviors; thus, their participation in school-based interventions and awareness programs is essential [49].

Policy-level changes, such as providing healthier food options in school cafeterias, are necessary to support these initiatives. Additionally, modifying the school environment to make nutritious choices more accessible and appealing can positively influence students' dietary habits. Peer support initiatives, such as structured peer groups, can further reinforce positive nutritional behaviors [49]. Beyond the school setting, community engagement programs—including collaborations with parents, local health professionals, and nutritionists—can provide continuous support through workshops, seminars, and community nutrition events. Empowering students as "nutritional ambassadors" can cultivate a culture of positive nutrition, wherein students lead initiatives, organize events, and serve as role models for their peers.

The integration of technology, such as mobile applications and online platforms, aligns with adolescents' evolving preferences and facilitates the dissemination of nutritional information and meal-planning resources. Furthermore, incorporating traditional Iranian cuisine into school meal plans—while ensuring nutritional balance—respects cultural food preferences while promoting healthier eating habits. While these proposed solutions are theoretically grounded, their effectiveness will ultimately depend on the lived experiences and perspectives of study participants. It is crucial to account for cultural, socioeconomic, and individual factors that may shape dietary behaviors and the efficacy of interventions [49]. Khuzestan, a hot and humid province in Iran, has distinct culinary traditions, with seafood and locally sourced vegetables forming a significant part of the diet. Additionally, the province's ethnic and religious diversity influences food preferences and nutritional patterns. Consequently, interventions should not rely on generic solutions but should instead be tailored to the specific characteristics and needs of the local population.

## Conclusion

A mixed-method study utilizing SCT constructs holds significant potential for identifying effective strategies to enhance the nutritional behaviors of Iranian female school students. However, the success of these interventions depends on their practical implementation and acceptance by students, parents, and school authorities. Given the multifaceted nature of nutritional behaviors, a holistic, context-specific approach is essential, encompassing educational, familial, policy-driven, environmental, and social dimensions. While the proposed interventions are theoretically sound, they remain hypotheses that require empirical validation. The distinctive cultural and socioeconomic landscape of Iranian secondary schools underscores the necessity of tailoring strategies to local contexts. Future research should prioritize the implementation and rigorous evaluation of these proposed solutions in real-world settings, ensuring continuous refinement to address emerging challenges. This dynamic process—from theory to practice—necessitates an iterative cycle of research, implementation, assessment, and adaptation. Through such systematic efforts, sustainable improvements in nutritional

behaviors among Iranian female school students can be achieved, ultimately contributing to both their immediate well-being and the establishment of lifelong healthy eating habits.

## Limitations and considerations for future research

First, the study's focus on female school students within Khuzestan province means the resulting 'solution box' will be highly context-specific. This will limit the direct generalizability of the intervention itself to other cultural or regional settings. However, we contend that the primary contribution is the **methodology**—a replicable, bottom-up framework for co-developing and prioritizing public health solutions—which is intentionally designed to be transferable and adaptable to other populations.

Second, the research will rely on self-reported data during both the initial qualitative exploration and the quantitative prioritization phases. This introduces a potential for social desirability or recall bias. To mitigate this, our facilitators will be thoroughly trained in building rapport and fostering a neutral environment to encourage candid responses from participants.

Finally, and most importantly, this protocol outlines the **development and prioritization** of an intervention, not the evaluation of its effectiveness. The design is not longitudinal and therefore cannot establish causal relationships between the 'solution box' and changes in health outcomes.

Consequently, these limitations clearly delineate the path for future research. The most critical next step will be to conduct a **longitudinal controlled trial** to rigorously evaluate the efficacy of the developed 'solution box' on measurable nutritional behaviors and clinical health indicators. Future studies could also apply our mixed-methods developmental framework to other health challenges or diverse populations, further validating its utility as a public health tool.

## Acknowledgments

We would like to express our sincere gratitude to all the Iranian female school who participated in this study. Their willingness to share their time and insights is invaluable to this research. For this research on the experiences of female school students, parents, and stakeholders, we will engage them as the general public.

## Author contributions

**Conceptualization:** Fatemeh Zarei.

**Data curation:** Fazlollah Ahmadi.

**Methodology:** Fatemeh Zarei, Fazlollah Ahmadi.

**Supervision:** Fatemeh Zarei, Fazlollah Ghofranipour.

**Validation:** Fazlollah Ahmadi.

**Writing – original draft:** Nasrin Mehrjoyan.

**Writing – review & editing:** Fatemeh Zarei, Fazlollah Ghofranipour, Fazlollah Ahmadi.

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
