## [Decision Letter · Decision Letter 0]

Dear Dr. Zarei,

Thank you for submitting your manuscript to PLOS ONE. After careful consideration, we feel that it has merit but does not fully meet PLOS ONE’s publication criteria as it currently stands. Therefore, we invite you to submit a revised version of the manuscript that addresses the points raised during the review process.

We look forward to receiving your revised manuscript.

Kind regards,

Hadi Ghasemi

Academic Editor

PLOS ONE

Journal Requirements:

Reviewers' comments:

Reviewer's Responses to Questions

**Comments to the Author**

1. Does the manuscript provide a valid rationale for the proposed study, with clearly identified and justified research questions?

Reviewer #1: Partly

Reviewer #2: Yes

2. Is the protocol technically sound and planned in a manner that will lead to a meaningful outcome and allow testing the stated hypotheses?

Reviewer #1: Partly

Reviewer #2: Yes

3. Is the methodology feasible and described in sufficient detail to allow the work to be replicable?

Reviewer #1: Yes

Reviewer #2: Yes

4. Have the authors described where all data underlying the findings will be made available when the study is complete?

Reviewer #1: No

Reviewer #2: No

5. Is the manuscript presented in an intelligible fashion and written in standard English?

Reviewer #1: No

Reviewer #2: Yes

You may also provide optional suggestions and comments to authors that they might find helpful in planning their study.

Reviewer #1: Development of a Proposed Solutions Box for Improving Nutritional Behaviors of Female 2 School Students: Protocol of a Mixed-Method Study

Research like this is worth pursuing and provides deep insight into preventive medicine but some major recommendations can be found below

Overall editing of the text is recommended to make the content structurally understandable and simplified for better readability. Such as shortening the sentences and improving transitions between ideas helps readers to get a better your point and understand the logic and rationale of your study.

Abstract:

The Aim part fails to properly state and justify the purpose and necessity of conducting the study, and reading this section provides very incomplete information. What exactly is meant by nutritional behaviors? What does a solution box mean and what does it include? How useful and sufficient can changing nutritional behaviors be?

In the method section, it is first mentioned that the study will be conducted in three stages and the first stage consists of two parts, but steps )two to three ( are not properly separated and explained. Following a consistent process in explaining the steps is very helpful in making each step more understandable and might help readability.

For example:

Stage 1: Qualitative Phase

Stage 2: Quantitative Phase

Stage 3: Solutions Optimization

Background: line 61 needs a reference.

Line 74-80 needs a reference?

Line 78 to 80, what do you mean by eating behavior? Is it a new factor to be described?

Line 83 to 85:These behaviors are associated with a high-risk 84 diet, including inadequate consumption of fruits and vegetables, physical inactivity, insufficient fiber intake, skipping breakfast, and inadequate milk and dairy product consumption,” physical inactivity is not a part of the high-risk diet!

Line 82 to 89 You can not connect global eating behavior prevalence with childhood obesity prevalence in Iran,. Please be more concise and determine what are you exactly trying to report.

Line 96 the authors again define eating behaviors with a recent reference, which is almost similar to what they had discussed in line 76. This repetition disrupts consistency in the text and may confuse the reader. It is better to adopt and follow a regular structure in writing the introduction, in such a way that the definitions of eating behaviors, eating habits, eating disorders, and problems are discussed with relevant and correct references, the meaning of eating behaviors is expressed separately in a way that its difference from other meanings is clear, and then regional and global statistics are discussed in a manner that is exactly in line with the definition.

Method:

Line133” The philosophical 133 foundation of this research is the pragmatism approach. In this approach, the advantages of both 134 quantitative and qualitative approaches are used to reach the goals of the research” Please make it clear, how can this approach benefit the study.

Line 130:be consistent with the terminology first you say qualitative study, then you talk about qualitative and quantitative approaches! “The mixed method has two 131 phases, and because of the subjectivity of the concept of nutritional behaviors, qualitative study is 132 considered the most suitable method for the first phase of this research. The philosophical 133 foundation of this research is the pragmatism approach. In this approach, the advantages of both 134 quantitative and qualitative approaches are used to reach the goals of the research”

Please clarify the difference between lines 147 and 150.

Please explain the solution box and how it fits within the study.

Line 168:Specify data collection and analyzing method”e research method using the directed content analysis approach using 169 Social Cognitive Theory constructs”

Line 187: Please provide the ethics approval committee code.

Line 222 needs a reference

Line 255. How do you choose the schools? What is the method of sampling to cover different zones?

Line 290 needs reference.

Discussion: considering food security (available and accessible) and its direct impact on dietary habits is crucial.

Reviewer #2: The comments are provided in the attached document. Please refer to the comments for a more comprehensive review of the manuscript.

**Do you want your identity to be public for this peer review?** For information about this choice, including consent withdrawal, please see our Privacy Policy

Reviewer #1: No

Reviewer #2: No

---

## [Author Response · Author response to Decision Letter 1]

21 Feb 2025

Response to Reviewers

Dear Reviewers,

We sincerely appreciate your valuable time and insightful comments on our manuscript. Your feedback has significantly improved the quality of our work. We have carefully addressed all your suggestions and revisions, ensuring that each point has been thoroughly considered.

Thank you for your constructive review and support.

Best regards

Corresponding Athours

The response letter to reviwer is attached in system

---

## [Decision Letter · Decision Letter 1]

Dear Dr. Zarei,

Thank you for submitting your manuscript to PLOS ONE. After careful consideration, we feel that it has merit but does not fully meet PLOS ONE’s publication criteria as it currently stands. Therefore, we invite you to submit a revised version of the manuscript that addresses the points raised during the review process.

We look forward to receiving your revised manuscript.

Kind regards,

Hadi Ghasemi

Academic Editor

PLOS ONE

Journal Requirements:

Reviewers' comments:

Reviewer's Responses to Questions

**Comments to the Author**

1. Does the manuscript provide a valid rationale for the proposed study, with clearly identified and justified research questions?

Reviewer #3: Yes

Reviewer #4: Yes

2. Is the protocol technically sound and planned in a manner that will lead to a meaningful outcome and allow testing the stated hypotheses?

Reviewer #3: Yes

Reviewer #4: Yes

3. Is the methodology feasible and described in sufficient detail to allow the work to be replicable?

Reviewer #3: Yes

Reviewer #4: Yes

4. Have the authors described where all data underlying the findings will be made available when the study is complete?

Reviewer #3: Yes

Reviewer #4: Yes

5. Is the manuscript presented in an intelligible fashion and written in standard English?

Reviewer #3: Yes

Reviewer #4: Yes

You may also provide optional suggestions and comments to authors that they might find helpful in planning their study.

Reviewer #3: While this article is well-written and informative, and the authors' efforts to provide solutions for improving the nutritional behaviors of female school students are appreciated, a different approach to presenting its unique contributions may be beneficial.

Aabstract:

1. The explanation of the aim should be more concise.

2. Keywords should be verified against MeSH standards.

Introduction:

3. The necessity of using a mixed method should be stated more clearly.

Methodology: The methods are adequately described and allow for the replication of the study.

4. It seems that a brief explanation of how the results are used in the application should also be given

Discussion: The discussion is well-written and analyzes the implications of the results in light of the existing literature.

5. However, it falls short in providing a section on limitations and explaining how the new findings contribute to the existing body of literature.

Reviewer #4: Congratulations for your research, I really enjoy reading it. I will recommended to continue with this interesting research line.

**Do you want your identity to be public for this peer review?** For information about this choice, including consent withdrawal, please see our Privacy Policy

Reviewer #3: No

Reviewer #4: No

---

## [Author Response · Author response to Decision Letter 2]

3 Jul 2025

#Reviewer 3 Response

Development of a Proposed Solutions Box for Improving Nutritional Behaviors of Female 2 School Students: Protocol of a Mixed-Method Study

Reviewer #3: While this article is well-written and informative, and the authors' efforts to provide solutions for improving the nutritional behaviors of female school students are appreciated, a different approach to presenting its unique contributions may be beneficial.

Thank you for your thoughtful feedback. We have carefully addressed your comments and made the necessary revisions, as detailed below.

Abstract:

1. The explanation of the aim should be more concise. We thank the reviewer for this valuable suggestion. We agree that our original aim statement was too detailed and included information better placed elsewhere in the manuscript. To address this, we have revised the aim to be a single, focused sentence. The additional details describing the components of the 'solution box' have been moved to the Methods section, and the broader rationale has been integrated into the Introduction.

The revised aim now reads:

'The aim of this study was to develop an evidence-based, context-specific ‘solution box’ to improve the nutritional behaviors of female school students in Khuzestan province

The explanation about the Solution Box mentioned in details in moved Methods section lines: 378-380 Done

2. Keywords should be verified against MeSH standards. We thank the reviewer for this valuable suggestion. To addressed this comment all keywords cheaked in Mesh AND the URL of each words was added for consideration

1- Nutritional behavior changed into “Nutrition” that verified in MeSH: https://www.ncbi.nlm.nih.gov/mesh/?term=Nutrition

2-Female school students changed into Student : https://www.ncbi.nlm.nih.gov/mesh/68013334

3- Feeding Behavior was added : https://www.ncbi.nlm.nih.gov/mesh/68005247

4- Qualitative Research

https://www.ncbi.nlm.nih.gov/mesh/68036301

Done

Introduction:

3. The necessity of using a mixed method should be stated more clearly. "We thank the reviewer for this crucial point. We acknowledge that the rationale for our mixed-methods approach was not sufficiently articulated in the Introduction. To address this, we have now added a dedicated paragraph at the end of the Introduction, just before the statement of the aim, to explicitly justify the use of a sequential exploratory mixed-methods design.

The following paragraph was added in introduction

"Addressing the complexity of nutritional behaviors in this specific cultural context requires a methodology that can both explore nuanced local perspectives and systematically prioritize interventions. A single-method approach would be insufficient; a purely qualitative study could identify potential solutions but would lack a structured mechanism for prioritization, while a purely quantitative study would risk imposing pre-conceived solutions that are not relevant to the lived experiences of students in Khuzestan. Therefore, this study adopts a sequential exploratory mixed-methods design. The initial qualitative phase is essential to explore and identify culturally-specific barriers and stakeholder-proposed solutions. Subsequently, the quantitative phase is necessary to systematically rank and prioritize these qualitatively-derived solutions, building consensus and ensuring feasibility. This integration allows the qualitative findings to directly inform the quantitative instrument, resulting in a final ‘solution box’ that is both empirically robust and contextually grounded." Done

Methodology: The methods are adequately described and allow for the replication of the study. Thanks , we appreciate this feedback

4. It seems that a brief explanation of how the results are used in the application should also be given We thank the reviewer for this excellent suggestion to clarify the practical application of our study's output. We agree that a more detailed description of the final 'solution box' and its use is needed.

To address this, we have expanded upon Stage 3 in the Methods section, renaming it c) Visualization of Solutions Box: NUTRIBOX.' This revised section now explicitly describes how the prioritized solutions (the 'results' of the first two stages) will be translated into a user-centered digital toolkit. We have detailed the intended end-users (students, parents, and school staff) and provided concrete examples of the resources that will be developed for each group. This clarifies the pathway from our research findings to a tangible, actionable tool for the community. So the following paragraph was added in lines: 396-413

The final stage of this study focuses on translating the research findings into a practical and sustainable application. The "results" from the quantitative prioritization phase (Stage 2) will directly inform the creation of the final 'solution box.' This will not be a static report but a dynamic, user-centered digital toolkit designed for practical implementation by key community members.

The application will be developed as a web-based platform accessible to students, parents, and school health staff. The content will be tailored to each audience:

• For Female Students: The platform will feature interactive modules, culturally-appropriate healthy recipes, goal-setting worksheets, and short, engaging videos based on the prioritized solutions.

• For Parents: It will offer downloadable resources, such as fact sheets on adolescent nutrition, guides for healthy family meals, and strategies to support their children's dietary habits, all grounded in the solutions co-developed with the community.

• For School Staff and Stakeholders: The toolkit will provide a clear action plan, implementation guides for school-wide initiatives (e.g., healthy canteen policies), and ready-to-use educational materials for classroom integration.

By structuring the application this way, the study ensures that the empirically prioritized solutions are not merely identified but are transformed into an actionable resource that empowers the entire school community to foster healthier nutritional behaviors. Done

Discussion: The discussion is well-written and analyzes the implications of the results in light of the existing literature. Thanks , we appreciate this feedback

5. However, it falls short in providing a section on limitations and explaining how the new findings contribute to the existing body of literature. We sincerely thank the reviewer for pointing out these important omissions. We agree that a discussion of the study's limitations and a clearer statement of its contribution to the literature are essential for a robust protocol. It was added in lines 478 496

Done

Reviewer #4: Congratulations for your research, I really enjoy reading it. I will recommended to continue with this interesting research line.

Manuscript Number PONE-D-24-52896R1

Dear Authors congratulation for your innovative research.

I enjoyed reading it. Thanks , we appreciate this feedback

---

## [Editor Report · Decision Letter 2]

Development of a Proposed Solutions Box for Improving Nutritional Behaviors of Female School Students: Protocol of a Mixed-Method Study

PONE-D-24-52896R2

Dear Dr. Zarei,

We’re pleased to inform you that your manuscript has been judged scientifically suitable for publication and will be formally accepted for publication once it meets all outstanding technical requirements.

Kind regards,

Hadi Ghasemi

Academic Editor

PLOS ONE
---

## [Editor Report · Acceptance letter]

PONE-D-24-52896R2

PLOS ONE

Dear Dr. Zarei,

I'm pleased to inform you that your manuscript has been deemed suitable for publication in PLOS ONE. Congratulations! Your manuscript is now being handed over to our production team.

Kind regards,

on behalf of

Dr. Hadi Ghasemi

Academic Editor

PLOS ONE